# Service Selection Using an Ensemble Meta-Learning Classifier for Students with Disabilities

Abdallah Namoun [1,*], Mohammad Ali Humayun [2], Oussama BenRhouma [1], Burhan Rashid Hussein [3], Ali Tufail [4], Abdullah Alshanqiti [1] and Waqas Nawaz [1]

1   Faculty of Computer and Information Systems, Islamic University of Madinah, Madinah 42351, Saudi Arabia
2   Department of Computer Science, Information Technology University, Lahore, Pakistan
3   Rennes University, Inria, Inserm IRISA UMR 6074, Empenn, ERL U 1228 Rennes, France
4   School of Digital Science, Universiti Brunei Darussalam, Tungku Link, Gadong BE1410, Brunei
*   Correspondence: a.namoun@iu.edu.sa

**Abstract:** Students with special needs should be empowered to use assistive technologies and services that suit their individual circumstances and environments to maximize their learning attainment. Fortunately, modern distributed computing paradigms, such as the Internet of Things (IoT), cloud computing, and mobile computing, provide ample opportunities to create and offer a multitude of digital assistive services and devices for people with disabilities. However, choosing the appropriate services from a pool of competing services while satisfying the unique requirements of disabled learners remains a challenging research endeavor. In this article, we propose an ensemble meta-learning model that ranks and selects the best IoT services while considering the diverse needs of disabled students within the educational context. We train and test our deep ensemble meta-learning model using two synthetically generated assistive services datasets. The first dataset incorporates 50,000 records representing the possible use of 12 learning activities, fulfilled by 60 distinct assistive services. The second dataset includes a range of 120,000 service ratings of seven quality features, including response, availability, successibility, latency, cost, quality of service, and accessibility. Our deep learning model uses an ensemble of multiple input learners fused using a meta-classification network shared by all the outputs representing individual assistive services. The model achieves significantly better results than traditional machine learning models (i.e., support vector machine and random forest) and a simple feed-forward neural network model without the ensemble technique. Furthermore, we extended our model to utilize the accessibility rating of services to suggest appropriate educational services for disabled learners. The empirical results show the acceptability of our assistive service recommender for learners with disabilities.

**Keywords:** service selection; disabled students; learners with disabilities; quality of service; ensemble method; deep learning; assistive technology; Internet of Things

## 1. Introduction

With the fast evolution of digital technologies and the development of convenient and affordable gadgets, humans' reliance on digital means to access various services is increasing at an alarming rate. Teaching and learning are no exception, where innovative teaching methodologies are progressively relying on technological tools [1–3]. The ever-increasing online learning content and readily available interactive tools have made learning significantly easier and widely accessible. In addition, heavy reliance on technological gadgets has made the new generation of students more inclined towards digital tools as compared to printed books and handwriting, as they are more comfortable using technology for learning. However, this growing dependence on technology in the education sector might potentially leave behind students with disabilities. Disabled learners might feel challenged using gadgets and tools that do not support accessibility and inclusion; for example, it

is more difficult for blind or visually impaired students to benefit effectively from online video lectures in their standard form (i.e., without video–audio descriptions and accessibility features). Hence, a parallel effort is required to develop assistive technologies that can make use of online learning content more conveniently for all forms of disabilities. Disabled students may have various impairments, such as visual, hearing, or physical mobility issues, which hinder their learning activities.

Recently, Internet of Things (IoT)-based automated environments have been developed to assist people with disabilities in different areas [4–6]. Some assistive tools and devices have been proposed to empower learning for students with disabilities [7–10]. However, no standard approach integrates all such services to benefit a wide range of disabilities. Furthermore, little research has focused on the analysis of behaviors of students with disabilities in terms of their preferences for certain services. Such analysis can be utilized by integrating platforms where relevant services can be systematically recommended to disabled users based on service ratings of similar users in comparable conditions.

This research develops an ensemble meta-learning approach that incorporates the learning requirements of students with disabilities and suggests appropriate IoT services that can assist disabled learners based on the nature of their disabilities and physical environments. Our model predicts how likely it is that students will use certain services depending on the type of disability and learning conditions. Moreover, the deep learning model predicts the accessibility ratings other users are likely to assign to the available IoT services. The suggested model can easily be integrated with intelligent recommendation systems to support teaching and learning in online learning environments that target students with disabilities. Learning recommendation systems can use the prediction of service usage and its ratings. In a nutshell, our main research contributions are summarized as follows:

(1) Two IoT datasets are generated containing diverse user attributes, assistive services for disabled learners, and user ratings. The lack of real datasets for educational scenarios motivated us to create synthetic datasets for students with disabilities. These datasets can be easily refined and extended for other researchers in the same field.

(2) An ensemble meta-learning prediction model is developed to recommend learning services for disabled students. The predictions are based on the user's disabilities type and various environmental attributes. Our model capitalizes on two key concepts, namely meta-learning and ensemble methods, to improve service recommendations.

(3) The proposed model predicts the ratings that disabled students are likely to assign to IoT services based on a myriad of features (i.e., response time, availability, throughput, successiblity, reliability, and latency).

The rest of the paper is organized into seven sections. Section 2 presents the recent research efforts regarding the application of the Internet of Things for the learning of students with disabilities. Section 3 describes the synthetic datasets developed for this research. Section 4 explains the experimental setup and the ensemble classification model proposed in this paper. Section 5 analyzes the results achieved using the proposed model. Section 6 discusses the significance of our ensemble model and its predictions. Finally, Section 7 concludes the research with the main findings and defines directions for future research.

## 2. Challenges of Learning for Students with Disabilities

The World Health Organization (WHO) estimates that 1.3 billion people in the world experience a significant disability [11] and a whopping 13% of the civilian population in the U.S.A. has some disability [12]. As many as 65 million school-age children suffer from a form of disability [13]. According to N.C.E.S., approximately 7.2 million students aged between 3 and 21 in the U.S.A. alone received accessibility services in 2020–2021 [13]. These statistics show that there is a dire need to help people with disabilities in performing various tasks. Disability can take various forms, for example, vision, hearing, thinking, learning, movement, mental health, remembering, communicating, and social relationships [14]. Smart technologies and services can be vital means of providing the required assistance

to people with special needs. In the context of our paper, support is offered to disabled students through the provision of an immersive teaching and learning environment.

Students with disabilities may face significant challenges in their learning journey due to a lack of assistive services. Such challenges can be categorized as general challenges that might have an impact on disabled students regardless of the disability type and specific challenges that are related to the type of disability a student might have. For example, social, emotional, and behavioral challenges apply to all students with disabilities [15], whereas the teaching of Braille and other related tools are only related to visually impaired students [16].

Assistive technologies and services play an influential role in helping learners with disabilities by creating an inclusive environment that facilitates the attainment of the required knowledge and skills in the desired domain [1]. Several studies were conducted to demonstrate the support that the Internet of Things (IoT) can provide to students with disabilities [7,17–26]. Figure 1 depicts the growing market size of assistive technologies developed to satisfy the wants of the disabled and elderly. The figure represents the future trends and the potential the technology holds to assist disabled users in various ways.

The wide range of IoT services and their variable quality make their selection, depending on user requirements and the context of use, a worthwhile challenge to tackle. Students with special needs have different requirements and constraints compared to fully abled students. However, most of the previous works in the service selection territory concentrated on addressing the needs of able-bodied learners. For instance, recent surveys on the use of computational intelligence in service composition and prediction of student performance [27,28] confirm the lack of intelligent solutions dedicated to serving disabled students. Moreover, the competing nature of services and their quality features (e.g., availability, latency, compatibility, etc.) require a robust machine-learning model that fuses assistive services to fulfill the learning goals of disabled learners in an optimal manner.

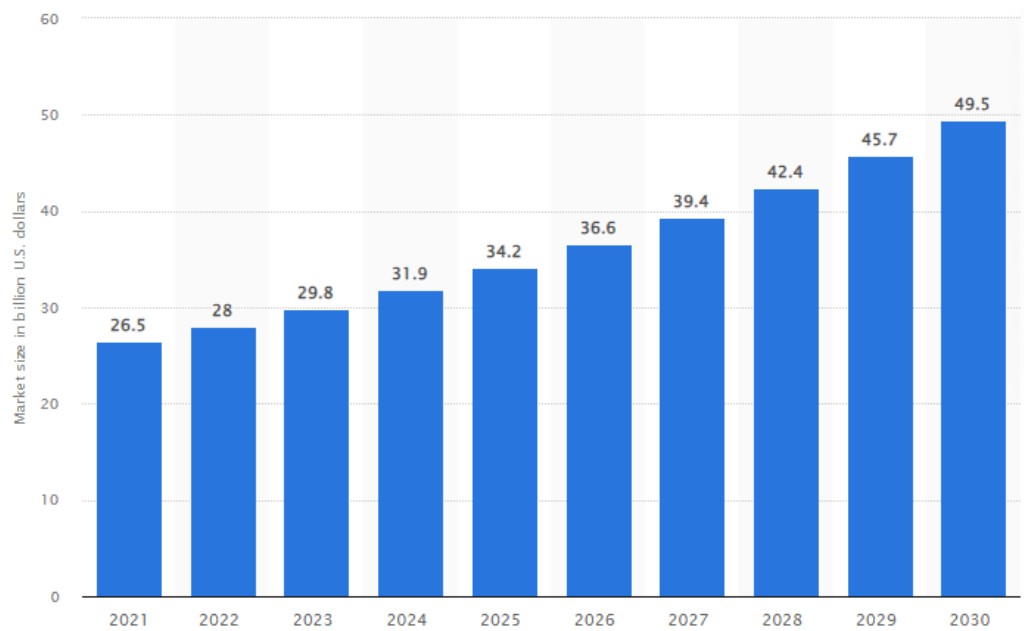

**Figure 1.** Size of the disabled and elderly assistive technologies market size between 2021 and 2030 (adopted from [29]).

The following is a summary of the major learning challenges students with disabilities face.

1.  General challenges: these types of challenges apply to disabled students regardless of the kind of disability. These include social, emotional, and behavioral difficulties that hinder the use of educational technologies. These challenges can be an artifact of the

contexts of user interaction, not the individuals themselves. For instance, a context where user privacy is infringed upon is often impeding to the user experience and poses social restrictions, causing frustration and poor experiences. This needs to be considered where providing a solution in the wrong context might lead to the failure of the solution itself.

For example, disabled students who find it hard to process information might suffer from low self-esteem and social withdrawal [15]. Additionally, barring students with disabilities from participating in school-related activities negatively impacts the psychological and social well-being of these students [16].

2.  Type-specific challenges: These types of challenges are related to the specific type of user impairment. For instance, the challenges a student with hearing impairments will face differ from those of a student with visual impairments.
3.  Service selection challenges: other challenges pertain to the assistive technology itself. This issue is more evident when several competing services are available from a wide range of sources, such as web services, mobile services, IoT services, cloud services, etc. It is not easy to automatically select the most appropriate services for students with disabilities while fulfilling their requirements and interaction context.

These challenges demonstrate the overarching need to develop intelligent frameworks that can search and select the most appropriate IoT services to accomplish the needs of a particular disabled student depending on their disability and learning context.

## 3. Related Work

### 3.1. Use of Internet of Things in Education

Considerable research has been conducted in the Internet of Things (IoT) domain, producing numerous applications to assist humans in performing routine tasks in their life [25]. IoT application areas span from smart cities to health services, waste management, traffic control, water and utility services, medical healthcare applications, and agriculture applications. In [23], the authors discussed the significance of IoT in collecting data for research via remote sensing from challenging environments, such as space. The applications of IoT are analyzed in hospitals for medication and treatment [21]. Equally, IoT-based environments have specifically helped people with disabilities in various ways. However, our study focuses on teaching and learning for students with special needs.

Interestingly, a handful of IoT initiatives are suggested in the literature to enhance the learning of disabled students, such as [7,9,17–20,22,24,26,30]. Heng et al. [20] suggested the use of IoT devices to help students with disabilities in their learning process. In [22], Hengeveld et al. developed a telepresence system for students along with a feature to monitor the rhythm and tone of teachers to analyze the requirements for student engagement. Similarly, Kassab et al. [17] summarized several studies that would help people with different disabilities, such as visual, hearing, and physical disabilities, to use IoT platforms. Kiryakova et al. [26] illustrated the value of interactive technology-based learning environments for toddlers. They proposed an environment where multiple digital interactive tools are utilized to improve language learning by toddlers with disabilities. The tools are designed to teach sign language to children with hearing impairments. They tested their model for four weeks and displayed significant improvements in the learning capabilities of students. Lenz et al. [19] developed an augmented reality software that identifies the name of the objects using computer vision and tags them with audio and text to teach students with impaired learning and comprehension capabilities. In another study [7], McRae et al. proposed an RFID-equipped library environment so that students with visual impairments can locate the books they search for with the help of RFID tags.

Moreover, Noor et al. [24] reviewed the benefits of wireless IoT devices in assisting the education of students with disabilities. They discussed the applications and challenges of using IoT devices in learning environments for disabled students. Similarly, Ullah et al. [18] highlighted the influence of IoT on teaching and learning by directly engaging with students

in real-time. They conducted a survey and found that 84% of the sample population indicated that technology helped them in their courses. Wambuaa et al. [9] presented a systematic literature review of the applications of IoT and models developed for educating disabled students. Services such as indoor thermal comforters and wheelchair systems were used for physical disabilities. In another study [30], Bluetooth beacon-based navigation was used to develop assistive walking for students with visual impairments. Real-time visual communication aids were developed for students with hearing impairments. An et al. [31] assembled wireless alerts, smart bracelets, and gloves for sign language recognition.

Lee, H. [32] discussed the trends and challenges of integrating IoT and assistive technologies in the lives of people with disabilities. The lack of clear guidelines on how to incorporate IoT into education remains a key challenge. In another study [33], Mishra et al. reviewed the importance of IoT in higher education to enhance interactive learning and teaching. They suggested that IoT could also assist disabled students in their learning. In a review article [34], Felicia et al. surveyed the Internet of Educational Things (aka IoEdT), emphasizing that IoEdT holds the future where they expect a better learning environment and learning processes that could benefit students with physical disabilities. In another review study [35], Algozani et al. discussed the impact of IoT on higher education, arguing that IoT overcomes the disadvantages of traditional online education and offers ample learning opportunities for students with disabilities. Elsaadany et al. [36] explored the use of IoT in education by experimenting with visually impaired students. They demonstrated that IoT can help students with special needs impart knowledge effectively. Similarly, Ding et al. [37] discussed the possibility of integrating IoT and virtual reality to assist students in physical education.

### 3.2. Machine Learning Models to Empower Disabled Students

Recently, researchers have employed machine learning in support of assistive services [38,39]. Poornappriya et al. [38] presented the application of machine learning to aid dyslexia prediction and support learning and cognitive disorders. Sharif et al. [39] proposed a machine learning-based approach that was applied in dyslexia scenarios to assist students with dyslectic needs with various teaching and learning activities. In another study [40], Namoun et al. investigated and proposed a two-phased machine-learning framework for context-aware service selection to help people with disabilities. By applying a scenario-based design technique, the researchers created a comprehensive disability ontology and a machine-learning framework for service selection for disabled people. However, this research aims to select learning services using an ensemble deep learning model for learners with disabilities.

Education service recommendation using deep learning was presented in [41], where the authors showed that collaborative filtering improves service recommendations' accuracy. Moreover, service selection using meta-learning is explored in these studies [42,43]. The authors suggest that using a meta-learner approach could help improve service ranking and selection. Service classification and recommendation using ensemble learning were also explored in other studies, e.g., refs. [44–46]. These studies demonstrate that the use of ensemble learning in recommendation and service selection systems is a viable and effective strategy. Ensemble learning is also applied in other interesting domains, showing an acceptable prediction performance [47]. However, the above studies did not address the requirements of disabled users, their unique context, and environments. Only a few recent studies have started to analyze the open challenges [27] and propose ML frameworks [40] for empowering the consumption of services and assistive technologies by people with disabilities in accessible modes.

### 3.3. Current Research Gaps

Our analysis of the related studies reveals that several research gaps remain unaddressed to date, as follows:

1. Most of the past works do not propose a predictive model to assist disabled students in choosing among assistive services depending on their needs and dynamic circumstances.
2. The role that an intelligent model can play in integrating various services has not been discussed in the context of supporting conducive learning environments for students with learning disabilities.
3. The context of disabled students has not been considered, including their social needs and accessibility requirements.
4. The application of deep learning for predicting IoT services to assist disabled students is not well-understood.
5. Machine learning solutions presented in previous studies, such as [33–35], do not address the user context problem, and their focus is narrowed to just a particular type of disability.

## 4. Development of IoT Assistive Services Datasets

Our research proposes a novel predictive model that helps to recommend assistive services to students with special needs. The proposed model predicts how likely a user with a specific disability in a particular environment would access a smart IoT service through their devices. Moreover, it anticipates the likelihood of obtaining a particular rating for a specific assistive service from a disabled student. For example, a user with a hearing disability is more likely to use a speech-to-text translation service (i.e., to be able to read the explanation) and will rate the service highly if it has a high availability and low latency. In the initial phase, we developed two synthetic datasets of IoT services to train and test the prediction model (the datasets are posted online on GitHub at: https://github.com/anamoun/servicesfordisabled, accessed on 10 April 2023). The rest of this section explains the rules employed to create the datasets.

The flowchart in Figure 2 depicts the creation of the synthetic datasets of assistive services. The datasets' creation assumes that disabled learners access various educational services using a myriad of digital devices. Important user characteristics considered in the interaction scenario include the type of disability, time of learning, location or environment, assistive technology used, and communication language. Below we explain the logic and rules that were implemented to derive the assistive services datasets.

For each disabled learner in the dataset, the time of learning is randomly selected from 'Morning', 'Afternoon', 'Evening', language of instruction from 'FirstLan', 'SecondLan', Physical Location from 'Public', 'Private', and disability type from 'Visual', 'Hearing', 'Physical', 'Blind', 'Deaf', 'Speech', 'None'. Next, the assistive device used by the student for accessing the educational materials is selected randomly from 'Desktop Computer', 'Tablet', 'Smart Phone', 'Smart T.V.', 'Smart Watch', and 'Voice Assistant'. However, based on the student's disability type, certain devices are omitted from the choice. For example, Lab P.C. is excluded from the possible devices for visually impaired or blind students or students with hand impairment. Similarly, the voice assistant is excluded for deaf users or students with speech or hearing impairments.

After selecting the assistive device, a subtask vector is generated for each student. The subtask vector contains 12 tasks corresponding to one of the 12 assistive services to achieve the learning outcomes. The set of assistive services includes user-location, class-notification, stream, speech-to-text, text-to-speech, speech-to-sign-language, sign-language-to-speech, subtitles, translate, customize-color, image-reader, and speech-commands. Each of the 12 elements in the subtask vector is initially set to 0 or 1 randomly, indicating if the disabled student has used that assistive service before.

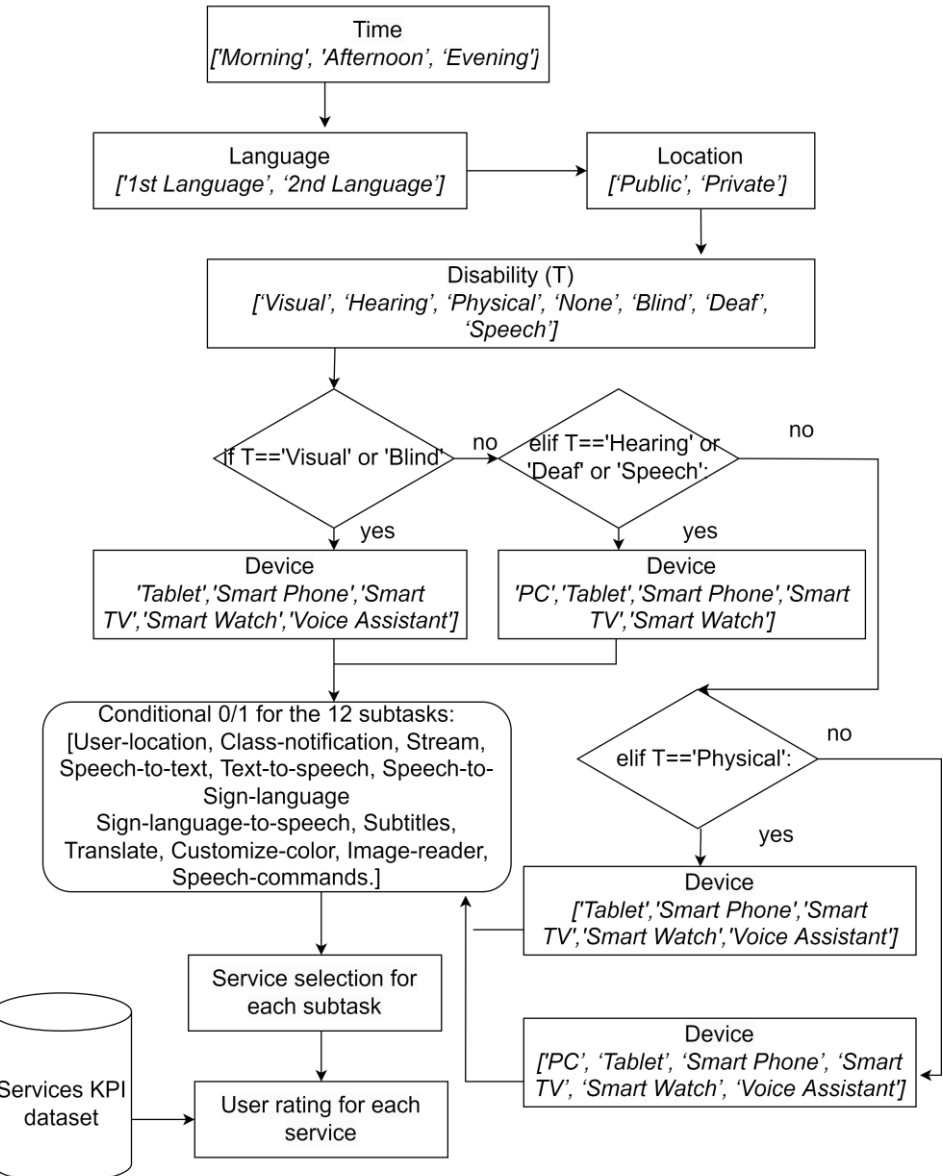

**Figure 2.** A flowchart outlining the logic of datasets' creation.

Finally, some subtasks are manually set to 0 or 1 to account for impossible scenarios of certain services. The streaming service is set to 0 if the user device is a 'Smart Watch' as smart watches cannot be used for video streaming. The speech-to-text service is impossible if the device is 'Voice Assistant', the disability is 'Speech' or 'Deafness', or the user location is 'Public'. Hence, these subtasks are set to 0 in such scenarios. Similarly, the text-to-speech service is set to 0 if the device is 'Voice Assistant' or 'Smart Watch' or Disability is 'Blind'. Likewise, if the device is 'Voice Assistant' or Device is 'Smart Watch' or the disability is not 'Deafness' or 'Hearing', then speech-to-sign-language and sign-language-to-speech services are set to 0. If the device is 'Voice Assistant' or 'Smart Watch' or the disability is 'Blindness', subtitles and text translation services are set to 0.

If the device is 'Voice Assistant' or disability is not 'Visual', customize-color service is set to 0, as only visually impaired users require this service, which is unavailable for voice assistant devices. Similarly, if the device is 'Voice Assistant' or 'Smart Watch' or the disability is not 'Blindness', then the image-reader is set to 0. Since users with a speech disability cannot use the speech-commands service, it is set to 0.

Since all services except user-location and class-notification depend on class streaming, if the streaming service is not used, all these services are set to 0. Moreover, since second language users must use subtitles and translation while streaming the class, if streaming is used and the language is 'SecondLan', the subtitles and translation services must be set to 1.

For each subtask, the following IoT services are selected, respectively: "Location service", Notifications, "Google meet", "Google A.S.R.", "Google TTS", "Sign language creation", "Sign language interpretation", "Subtitles", "Machine translation", "Apple resolution", "Image caption", "Alexa".

After the IoT services are determined, an Internet QoS performance dataset is used to generate user ratings of service accessibility. The IoT service rating is set based on the following features:

- Response time;
- Availability;
- Throughput;
- Successibility;
- Reliability;
- Latency.

Response time indicates the average round trip time for the request to each assistive service. Service availability indicates the average chance of the assistive service being operational each time it is requested. Throughput indicates the maximum data rate a service can offer. Successibility refers to the ratio of successful requests for the assistive service. Reliability indicates the average probability of a service delivering the required output. Finally, service latency is the time delay incurred for the request to reach the service.

All of the above-mentioned values were averaged and translated into a QoS rating for each assistive service. The QoS rating is added with a random noise fluctuation to incorporate human behavioral randomness and rounded off, as an integer, to eventually obtain a user accessibility rating between 1 = least accessible service and 5 = highest accessible service. If a particular student does not use a service, their rating is set to 0. All the steps mentioned above are repeated for each of the 3000 disabled students in the dataset.

## 5. The Proposed Ensemble-Based Deep Learning Model

Our suggested service prediction model takes features such as the time of the study, student location, type of device used to access the learning materials, and the disability of the student as inputs. Based on these inputs, two different classification models have been developed incrementally. The first model predicts the learning services a particular student is expected to utilize. In contrast, the second model predicts the student rating based on their experience of using the assistive services. While deploying for a real learning platform, disabled learners can manually submit their locations, public or private, their languages of instruction, and the specifics of their disabilities. Alternatively, a smart learning tool can automatically detect the time and device of the learner.

The overall experimental setup is outlined in Figure 3. Initially, a synthetic dataset was generated and split into training and test partitions. The training partition and service rating labels are used to train the proposed machine learning model, while the test partition is used to test the model in terms of predicting service rating labels.

The proposed model (see Figure 4) is based on an ensemble of multiple feed-forward learners fused using meta-learners that feed multiple output layers simultaneously. Each input for the meta-learner comprises one hot vector representing time, language, location, disability, and device. The different output layers corresponding to one of the service labels share the hidden layer or the meta learner. The output layers comprise two units for the binary service usage classification and six units for the service rating prediction. The model uses the 'Adam' optimizer for training and ties to minimize the cross-entropy loss. The architecture of the proposed model is illustrated in Figure 4.

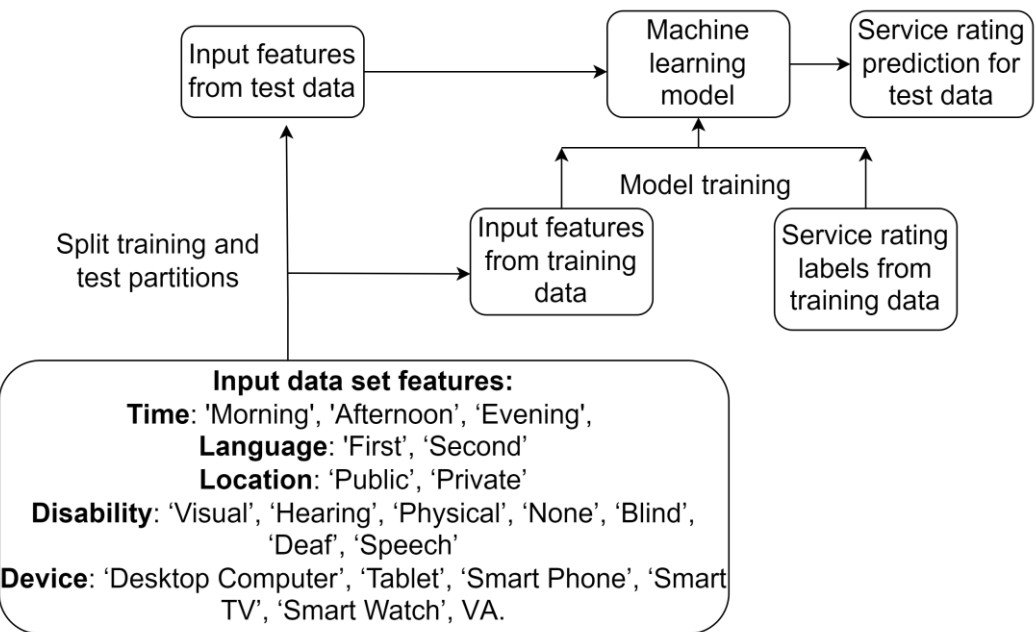

**Figure 3.** The experimental setup strategy.

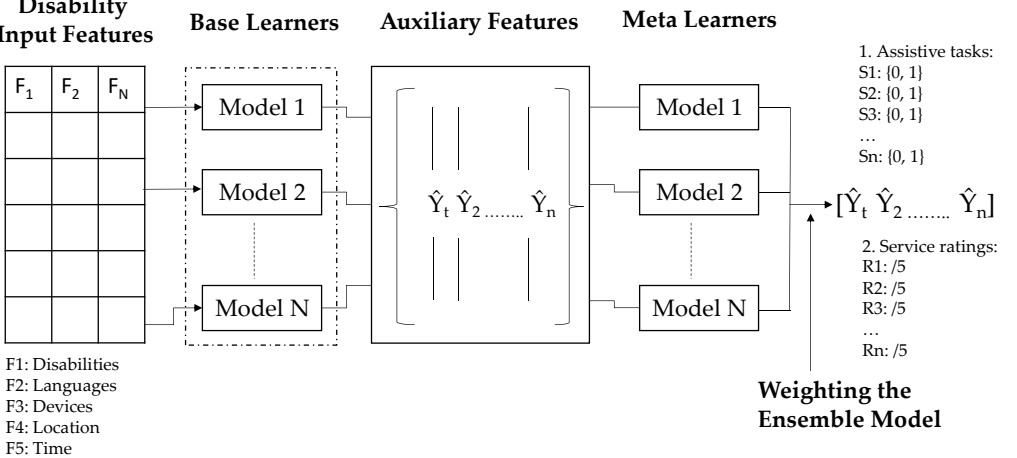

**Figure 4.** The proposed ensemble meta-learning architecture of our disability service prediction model (F: learning feature, S: assistive task/service, R: service rating on a 5-point scale).

Ensemble learning models aggregate multiple individual classifiers utilizing their diversity in capturing the input features. The individual models for the model proposed in this paper are based on different input features and try to learn the relationship between the service rating labels and these features separately.

Multiple aggregation methods, such as weighted averaging, simple averaging, or summation, can be used to merge the individual classifiers. However, recently, meta-learning has been more successful for ensemble models. Meta learners stack an additional classifier to merge the outputs of the individual classifiers. The meta-learning classifier automatically learns the emphasis for each of the individual classifiers feeding it. The overall network is also called a stacked network and is trained end-to-end using the training dataset. The proposed model utilizes meta-learning to merge the individual feature classifiers with additional hidden layers that learn while training the weights for combining the individual classifiers. The training process emphasizes the input classifiers that significantly impact the service rating prediction output.

The proposed model needs to simultaneously predict the labels for multiple services. Hence, the model is designed using multiple feed forward neural networks that predict the class labels in parallel using the input features. During training, each classifier fits class labels for a separate service category. Similarly, the classifiers infer the class labels for different services during the testing phase. In contrast to a single end-to-end feed forward neural network, the proposed model ensembles multiple stand-alone neural networks to make predictions. The models are trained separately, and their predictions are combined to compute the final performance measures.

The proposed stacked model needs to predict the labels for multiple services simultaneously. Hence, the meta-learning layers feed multiple output feed-forward layers in parallel. The models are trained separately, and their predictions are combined to compute the final performance measures.

All input features, i.e., time, language, location, disability, and device are separately converted to a one-hot vector. Time is represented as a one-hot vector with three dimensions, each indicating morning, evening, or afternoon with one. Language is converted into two-dimensional input, each indicating Arabic or Non-Arabic. Two-dimensional vectors also indicate the location to represent public or private spaces. A seven-dimensional vector, each indicating hearing, blind, physical visual deaf, speech, or no disability, represents the disability type. The device is represented by five-dimensional vectors, each dimension indicating tablet, smart watch, smart phone, smart T.V., or voice assistant.

The input layers represent one-hot vectors for each of the student features, i.e., disability, device location, time, and language. Each of these input layers feeds an individual learner comprising two dense hidden layers, each with linear activations and 12 neurons. The individual learners are then merged using a shared meta-learning model with three dense layers, each comprising 100 units with rectified linear activations. The shared meta-learner feeds 12 individual output models with dense Softmax layers. The Softmax output layers comprise two neurons for binary classification of service usage and six neurons for the service rating task, whereby five out of the six neurons represent the rating level for the service, whilst the sixth one represents service not being used.

The model is optimized using Adamax optimizer to minimize categorical cross and entropy loss. Figure 4 illustrates the ensemble neural network architecture used as the machine learning model. A dense layer with linear activations is used as the common hidden layer, while separate dense layers with softmax activations are used as output layers.

A total of 70% of the data was used to train the ensemble model, while the remaining 30% was retained to test the model. The prediction of the output layer with the highest probability is selected as the target student rating. The performance was evaluated by comparing it with the true labels within the dataset in terms of accuracy and F-score. Our meta-learning model is implemented in Keras.

Since our task involves a small number of input features for training classification models, a single trained model may not generalize well when deployed in production. Hence, our proposed method involves training multiple learners (e.g., heterogeneous models) to further improve the generalization of unseen data. These learners can be viewed as a set of feature extractors where predicted outputs are then combined (in the form of auxiliary features) and used as input features to train the meta-learners. All models' hyperparameters are optimized using a Bayesian approach, as depicted in Algorithm 1.

---

**Algorithm 1:** Pseudocode for the proposed ensemble meta-learning approach

---

**Input:** a training_dataset with student characteristics as input features, base_models,
meta_learner, Bayesian_optimization_algorithm, while tasks used and service ratings as
target labels
**Output:** ensemble_model with trained weights
*1. Initialize: list of base models and their predictions*
*Taking the student specific features as input and feeding its output as auxiliary features to the meta learners*
*2. For each base model in base_models:*
*    a. Optimize hyper parameters of the base model using Bayesian_optimization_algorithm on training_data*
*    b. Train the optimized base model on training_data*
*3. Combine the predictions of base models into auxiliary features*
*4. Train meta_learner using auxiliary features as input and the tasks usage along with the service ratings as separate outputs*
*5. Store the meta_learner's weights and predictions on validation_data*
*6. Repeat the steps until stopping criterion is met:*
*    a. Train meta_learner on validation_data using auxiliary features as input*
*    b. Store the meta_learner's weights and predictions on validation_data*
*7. Return ensemble_model consisting of the trained base learners and the updated meta learner*

---

## 6. Results

The prediction of the ensemble meta-learning model is evaluated in terms of primary performance metrics, including accuracy, and f-score, for both the service usage likelihood and service ratings by disabled learners. The results are compared in terms of classification accuracy and f-score. The classification accuracy represents the number of accurate predictions out of the total predictions. The f-score represents the harmonic mean of precision and recall. The harmonic mean is used to average the ratios. The precision indicates the number of actual protections out of the total predictions for a particular class. At the same time, recall represents the number of accurate predictions out of all the samples of that class. Weighted averages of precision and recall across all categories are computed to obtain the average metrics, which are then used to calculate the harmonic mean to represent the f-score. The accuracy and f-score for binary usage prediction of each service are reported separately in Table 1. The average accuracy across the 12 assistive tasks is 84.91%.

**Table 1.** Task classification scores.

| Task | Assistive Service | Accuracy | F-Score |
|---|---|---|---|
| Identify user location | Location service | 0.858 | 0.879 |
| Notify user about class | Notifications | 0.846 | 0.832 |
| Live stream the classroom | Google Meet | 0.86 | 0.832 |
| Speech to text | Google ASR | 0.845 | 0.841 |
| Text to speech | Google TTS | 0.831 | 0.864 |
| Speech to sign language | Sign language creation | 0.857 | 0.908 |
| Sign language to speech | Sign language interpretation | 0.844 | 0.873 |
| Show subtitles or captions on screen | Subtitles | 0.854 | 0.912 |
| Translate text from Arabic to English | Machine translation | 0.85 | 0.864 |
| Customize screen color contrast | Apple resolution | 0.83 | 0.883 |
| Image text reader | Image caption | 0.857 | 0.903 |
| Speech commands | Amazon Alexa | 0.858 | 0.853 |

Table 2 reports on the classification accuracy for predicting the user rating for each IoT service. The scenario of a service not being used is also included in this second task, with a 0-rating value reserved for that. The average accuracy across 12 services with six outputs to each service is around 58.47%.

**Table 2.** Assistive service rating classification scores.

| Task | Assistive Service | Accuracy | F-Score |
|------|-------------------|----------|---------|
| Identify user location | Location service | 0.581 | 0.602 |
| Notify user about class | Notifications | 0.561 | 0.561 |
| Live stream the classroom | Google Meet | 0.596 | 0.585 |
| Speech to text | Google ASR | 0.575 | 0.578 |
| Text to speech | Google TTS | 0.598 | 0.591 |
| Speech to sign language | Sign language creation | 0.594 | 0.584 |
| Sign language to speech | Sign language interpretation | 0.563 | 0.591 |
| Show subtitles or captions on screen | Subtitles | 0.592 | 0.59 |
| Translate text from Arabic to English | Machine translation | 0.577 | 0.568 |
| Customize screen color contrast | Apple resolution | 0.6 | 0.606 |
| Image text reader | Image caption | 0.576 | 0.566 |
| Speech commands | Amazon Alexa | 0.604 | 0.569 |

The performance scores for the task of service rating prediction are in a lower range because the task is more complex and involves more randomness. All the classification scores lie in the range of 56 to 60%. However, the performance scores are still quite significant as the classification scores are above the chance level, around 20% for a task with five possible outcomes. Moreover, the classification accuracy indicates that the rating for the speech-to-text service is relatively easier to predict. This is logical, as students with hearing or speaking disabilities are likely to rate such services higher.

Figures 5 and 6 compare the performance scores across the assistive services, as bar charts, for both classification tasks. Figure 7 indicates the optimization curves for the training of all the output layers corresponding to each of the indicated services.

The end-to-end optimization process aims to minimize the overall loss of the network. The overall loss is the sum of the optimization losses for each output corresponding to one of the 12 services. The optimization curves illustrate the loss minimization for each service output. The overall optimization curve is the sum of the individual curves. Similarly, the performance scores indicate each service's class and optimization curves. Most of the services achieve similar results with slight variations.

The classification results are significant in terms of binary prediction of tasks as any recommendation system can utilize this classification model and consequently predict if a disabled person will be interested in a particular assistive service. Recommending relevant IoT services to users will build their trust in any platform. Similarly, the prediction model for user rating can also be used to analyze IoT-enabled platforms with highly rated services for students with disabilities.

For comparison with the benchmark, the same classification task was implemented using a simple feed-forward neural network, which achieves 82% accuracy for the binary classification and around 36% for service prediction, which is significantly lower than the results achieved by our proposed ensemble learning architecture. The results were also compared with shallow machine learning models, specifically support vector machine (SVM) and random forest. The results achieved using the traditional machine learning

models are significantly lower than those achieved with the deep learning models, with SVM reporting 73.67% and 24.32, while random forests resulted in 71.72% and 20.58% for the binary classification and service rating prediction, respectively.

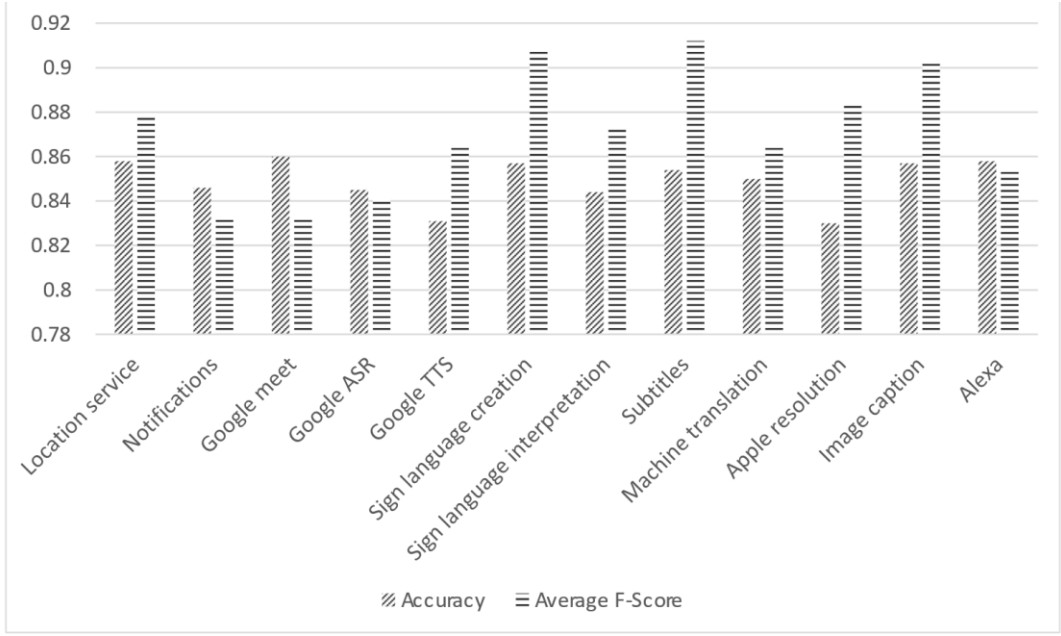

**Figure 5.** Binary classification scores (accuracy and f-score) of the assistive tasks.

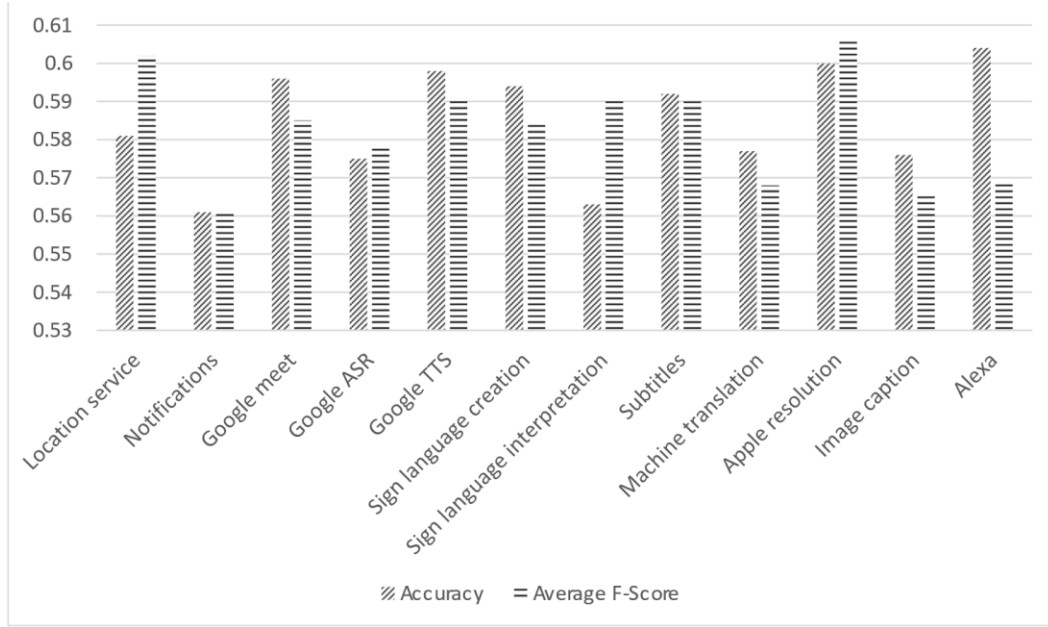

**Figure 6.** Service rating (of accessibility) prediction scores (accuracy and f-score).

The proposed ensemble model outperforms a single feedforward model in terms of accuracy as the ensemble approach helps the model learn complex relationships between features and labels of multiple services. Moreover, the ensemble training approach mitigates overfitting by reducing the impact of irrelevant features on the final predictions due to the simultaneous training of individual models. Figure 8 summarizes the comparison results of the prediction accuracy produced by our ensemble method against other simple ML architectures.

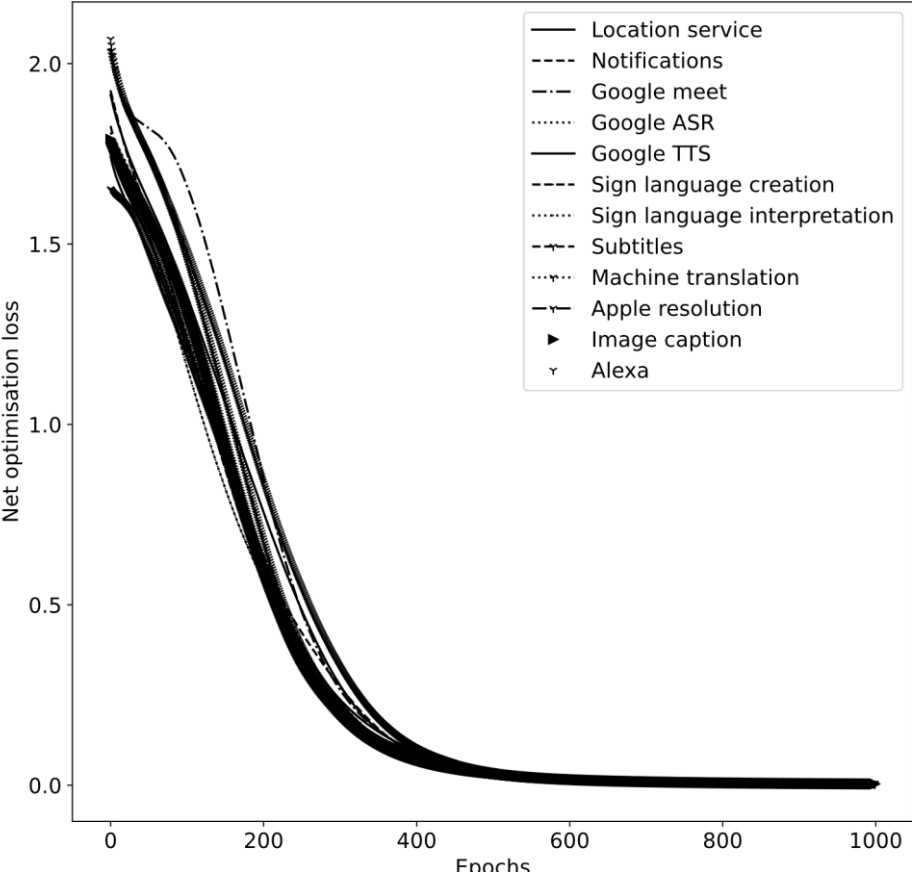

**Figure 7.** Optimization curves of the assistive services.

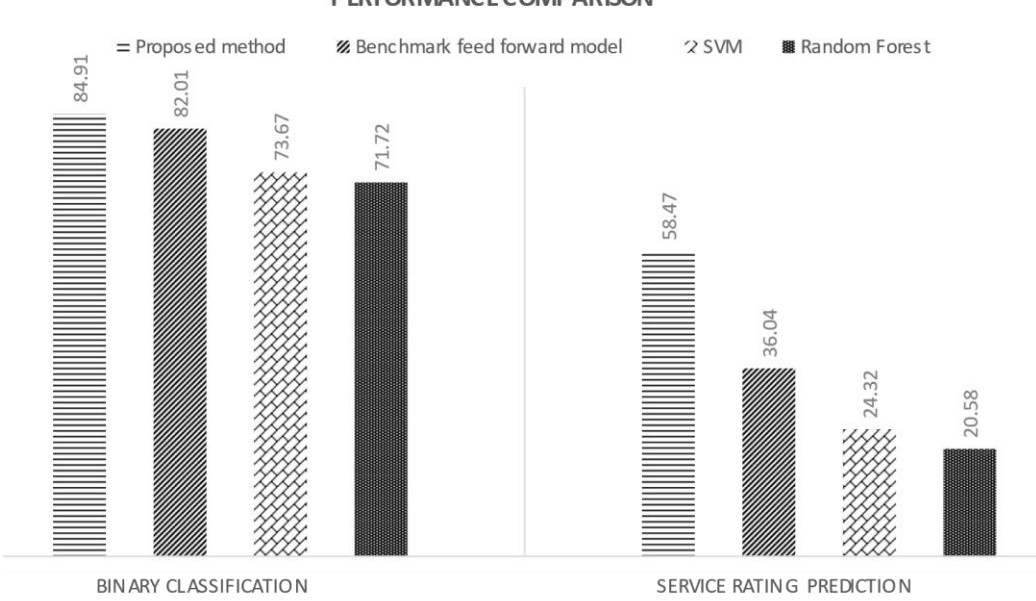

**Figure 8.** Benchmark accuracy performance results (our ensemble meta-learning model vs. simple feed-forward neural network, support vector machine, and random forest) of assistive tasks and service rating classification.

## 7. Discussion of Key Findings and Limitations

The proposed prediction model was tested in two scenarios: (a) for binary classification to predict if a disabled user will use a service or not; and (b) for multiple category classification to predict the user rating for the assistive service. The results for binary classification for service usage are encouraging, and the model correctly identifies whether a user will use a service. This result motivates the online learning tools to incorporate our prediction model as part of their architecture to recommend assistive services to students based on their input parameters. The correct recommendations put forward by the model will, in turn, build trust for the student in the learning platform. The positive impact on the user created by the proper recommendations develops a friendly and trustworthy human–computer interface. Consequently, the user will more likely rate the service and the learning tool highly, which will also affect the secondary classification task.

However, the second model to predict user ratings for the current dataset needs to report satisfactory accuracy, although it is way above the chance level of random classification for five categories. The lower accuracy for service rating can be attributed to an increased amount of randomness in the dataset for user rating of the service compared to service usage. Service usage is more deterministic than service rating as usage is more tightly bound to user parameters such as disability type, usage device, and user location.

A major limitation of this research is the synthetic nature of the datasets that we used in our research. This decision was made because of the scarcity of this type of data. It is noteworthy that the dataset developed and used in this research has a specific amount of deterministic algorithm and randomness, as explained earlier in the dataset section. The synthetic nature of the data is also visible in the different performance scores for both classification tasks. It may be possible that testing the proposed model using real datasets might yield different performance outcomes. Another limitation concerns our weak understanding of the mental models of people with disabilities and how they utilize assistive services. Further user studies are advocated to decipher the ways in which disabled learners interact with smart educational services, with opportunities to fuse those services in different contexts to achieve complex goals. Examples of user studies that established a strong theoretical understanding of the mental models of users who aggregate software services include [48,49]. Both studies also highlight the challenges of understanding user mental models in the context of service development.

## 8. Concluding Remarks and Future Work

With technological advancements, modern education systems rely more on online and digital learning tools. However, the needs of students with disabilities must be catered for by such digital learning platforms. Our research addresses the problem of using Internet of Things-based environments by students with disabilities to enhance their learning experiences. IoT has been used recently for intelligent environments in many areas; however, to the best of our knowledge, this study's application of deep learning for predicting IoT services to assist disability users is the first effort of its kind.

This paper proposed a deep learning-based prediction model which predicts the services a learning environment must suggest to students with different disabilities. The model predicts whether a user will use a service or not, along with the accessibility rating for the service on a 5-point Likert scale (least accessible to highly accessible). The proposed ensemble neural network model achieved an overall accuracy of 84.91% for the binary classification of using a service and an overall accuracy of just around 58.47% for predicting service ratings by users with disabilities. Our prediction scores are superior to those achieved with a simple neural network and traditional machine learning models (e.g., SVM and Randon Forest), showing the importance of ensemble and meta-learning in ameliorating service predictions for people with special needs.

The predictions of our model can be used as a module in learning recommendation systems for suggesting assistive services to students with disabilities in virtual environ-

ments. Developing an intelligent platform with these integrated IoT services and designing an end-to-end recommendation system is the logical way forward for research.

**Author Contributions:** Conceptualization, A.N.; methodology, A.N., M.A.H. and B.R.H.; formal analysis, M.A.H. and A.N.; investigation, A.N. and M.A.H.; data curation, M.A.H. and A.N.; writing—original draft preparation, A.N. and M.A.H.; writing—review and editing, A.N., M.A.H., A.T., O.B., A.A. and W.N.; supervision, A.N.; project administration, A.N.; funding acquisition, A.N. All authors have read and agreed to the published version of the manuscript.

**Funding:** This research was funded by the Deputyship for Research and Innovation, the Ministry of Education in Saudi Arabia, through project number 13/20.

**Institutional Review Board Statement:** Not applicable.

**Informed Consent Statement:** Not applicable.

**Data Availability Statement:** Our research data are publicly available at: https://github.com/anamoun/servicesfordisabled (accessed on 10 April 2023).

**Conflicts of Interest:** The authors declare no conflict of interest.

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
