# Peer review of "Service Selection Using an Ensemble Meta-Learning Classifier for Students with Disabilities"

_mti, doi:10.3390/mti7050042_

Round 1

Reviewer 1 Report (Previous Reviewer 2)

Having read the revised final version according to the reviewers' comments, there is a clear improvement in both the readability of the text and the graphic aspect, so I have no hesitation in suggesting the publication of the paper in its present form.

Author Response

Thank you for your positive feedback. There are no major comments from the reviewer. 

Reviewer 2 Report (Previous Reviewer 3)

Thank you for the revision.

Author Response

Thank you for your positive feedback. There are no major comments from the reviewer. 

Reviewer 3 Report (New Reviewer)

Abstract is informative and well-wrriten - please provide some figures next to "significantly better" assessment.

Introduction is well written and well-structured. The referencing is messy in the begining, please- dont assign 6 sources to a very general and well-known sentence - check the referencing through the whole section - please avoid referencing very general ideas with multiple sources at once. If you choose to show the background and theoretical support - mention the research on the matter and put sources - not just the idea.

Section 2 is good, statisrtcs is fresh  - of 2023. The figure 1 is a photo and the quality is not good  - please check. The authors successfully and clearly summarized the challenges - this is a good contribution of the section 

Literature review section is ok and proves a solid theoretical background of the team. Subsections 3.2 and 3.3 add the value to the section with a structured identified research gaps. This makes the section logically coherent and proves the originality of the research problem. 

Sections 4 and 5 describle the models in details, they are easy to follow and well-structured. Results section is informative and provides a good analysis of the outcomes obtained in the experiment. Limitations section as well as the duscussion provide a clear view of the authors on the focus of the future reseacrh

Author Response

Thank you for your suggestions. Below, we addressed the raised concerns:

  1. We have revised the references as per the suggested feedback. 
  2. We have updated Figure 1 with a better resolution. 

Kindly check the revised manuscript. 

This manuscript is a resubmission of an earlier submission. The following is a list of the peer review reports and author responses from that submission.

Round 1

Reviewer 1 Report

They propose a deep learning-based model to predict and evaluate learning services for students with specific disabilities.

Comments:

  • It is unclear how all references to the Internet of Things make a real contribution to the work.

  • The results are borderline modest.

  • Has an evaluation been done with data from real users? Because the stated semi-synthetic data seems completely synthetic.

  • Line 77, the third contribution does not seem a real contribution, especially in light of the rating results.

  • Line 80, check sessions’ references.

  • At the end of the reading of Section 2, the challenges are not clear to the reader, this part should be deeply revised.

  • Introduce acronyms with full terminology in the first instance so readers know what they mean, e.g., NCES? ASR? TTS? QoS?

  • Line 110, not all references in the range [11-21] are related to disabilities.

  • Section 3 seems just a long list, for many of them, the relationship to the proposed work is not clear.

  • Lines 197-200, an example can be useful for the reader.

  • Line 202, semi-synthetic should be full-synthetic.

  • In Figure 2, should all the "Device" boxes point to the “Condition 0/1 for the 12 subtasks” box?

  • Line 269, “The QoS rating is added with a randomly assigned user rating by each user to eventually obtain a user accessibility rating between 1 to 5 as an integer.”? What does “randomly assigned user” mean?

  • Line 272, “each of the 10000 users” real users or synthetic users?

  • Figure 3 is not clear.

  • In lines 289-296, the presentation of the architecture is unclear and there are strong doubts as to whether it is really meta-learning.

  • Line 330, 12 output layers or nodes?

  • Line 336, Figure 3?

  • Figure 6, the vertical axis of the graphs and a normalized scale to quickly understand the quality of the results are missing. Only the columns are misleading.

  • Line 372, “are considerably above the chance level”, really?

  • In References, the formatting must be checked and the conference/magazine information of many references is missing (e.g., 5, 8, 10, ...).

Author Response

First, we would like to thank the reviewer for their valuable comments. We made a substantial revision to the article in order to satisfy all suggestions. In the manuscript, the changes are highlighted in yellow and word track changes feature.

A major revision was done to the study and manuscript. 

Reviewer 2 Report

Congratulations to  the Authors for this great  article.

1.               Data - The fact that the authors - generously - have provided the data on which they based part of their work, allows  its use by other researchers in complementary or related investigations. Knowing that the data is partially synthetic does not reduce its merit, and the relevance of its contribution   to knowledge in this area. Specifically, this data is very useful for testing automatic learning algorithms.

2.              References - It would be interesting to add additional references covering the areas of deep-learning, meta-learners and ensemble methods.

3.              Figures 6 and 8. Printed versions of legends in figures 6 and 8 are difficult to read.  Perhaps another choice of colors and size of the symbols would improve the reading.

4.              Table 2. Please explain – perhaps in the legend -  the meaning of "Acc." And "F-Score"

Author Response

(The authors gave the same response as above.)

Reviewer 3 Report

1.       “Authors in [11]”. Always name authors in such situations, e.g. Kassab and Darabkh [11]. Correct all the similar situations.

2.       “However, the focus of our research is to provide learning services using deep learning for the disabled students”. Why do you stress the deep learning, but not provide the reasons why to use the deep learning?

3.       The review of related work must be critical. Now, it is just presentation of the contents of the related works.

4.        “the focus of our research is to provide learning services using deep learning for the disabled students”. Do not write the generic statements. Define your used method and provide a reasoning.

5.        “Figure 3 illustrates the Neural Network architecture”. False reference.

6.       We find plural – “hidden layers” in Fig. 4. However, we observe the singular “hidden layer” in the explanatory text.

7.       Figures 5 and 6 compares” – English grammar

8.       The axies in Fig. 7 have no measurement units.

Author Response

(The authors gave the same response as above.)

Round 2

Reviewer 1 Report

There are still unresolved problem with the manuscript:

  • The model is not well-described. What are these individual classifiers that form the final model for the ensemble learning? Are the layers of the neural network, if so it is not an ensemble learning.

  • The IoT is still not motivated, perhaps it is not necessary.

  • What is the added value brought by [51] [52]?

  • Has an evaluation been made on real data, not the synthetic one?

  • Some figures are of low quality.